# Discrimination of Classical and Atypical BSE by a Distinct Immunohistochemical PrP^Sc^ Profile

**DOI:** 10.3390/pathogens12020353

**Published:** 2023-02-20

**Authors:** Christine Fast, Catherine Graham, Martin Kaatz, Kristina Santiago-Mateo, Tammy Kaatz, Kendra MacPherson, Anne Balkema-Buschmann, Ute Ziegler, Martin H. Groschup, Stefanie Czub

**Affiliations:** 1Friedrich-Loeffler Institut/INEID, 17493 Insel Riems, Germany; 2Canadian Food Inspection Agency, Lethbridge, AB T1J 3Z4, Canada; 3Faculty of Veterinary Medicine, University of Calgary, Calgary, AB T2N 1N4, Canada

**Keywords:** classical BSE, atypical BSE, H-type BSE, L-type BSE, PrP^Sc^-profile, pathological prion protein, BSE discrimination

## Abstract

Bovine spongiform encephalopathy (BSE) belongs to the group of transmissible spongiform encephalopathies and is associated with the accumulation of a pathological isoform of the host-encoded glycoprotein, designated prion protein (PrP^Sc^). Classical BSE (C-type) and two atypical BSE forms (L- and H-type) are known, and can be discriminated by biochemical characteristics. The goal of our study was to identify type-specific PrP^Sc^ profiles by using Immunohistochemistry. In our study, brain samples from 21 cattle, intracerebrally inoculated with C-, H-, and L-type BSE, were used. In addition, the corresponding samples from three orally C-type BSE infected animals were also included. From all animals, a lesion and PrP^Sc^-profiles of six brain regions were determined. The lesion profile and the neuroanatomical distribution of PrP^Sc^ was highly consistent between the groups, but the immunohistochemical analysis revealed a distinct PrP^Sc^ profile for the different BSE-types, which included both the topographic and cellular pattern of PrP^Sc^. This qualitative and quantitative analysis of PrP^Sc^ affected structures sheds new light into the pathogenesis of the different BSE types. Furthermore, immunohistochemical characterization is supported as an additional diagnostic tool in BSE surveillance programs, especially when only formalin-fixed tissue samples are available.

## 1. Introduction

Transmissible spongiform encephalopathies (TSE) are a complex group of chronic fatal neurodegenerative disorders, which include genetic, infectious, and sporadic diseases in several species, including humans. In all TSEs the host-encoded cellular prion protein (PrP^C^) is converted into a stable, partially protease-resistant pathological isoform (PrP^Sc^), which accumulates over time and is the diagnostic disease marker detectable by immunohistochemistry (IHC) and biochemically [1].

In contrast to scrapie in sheep, the prototype of all TSE, which has been known since more than 270 years [2], bovine spongiform encephalopathy (BSE), was not diagnosed until 1986 [3]. This so-called classical BSE (“C-type”) originated from oral uptake of infectious feedstuff and, during the BSE epidemic, affected more than 185.000 clinically and fatally diseased cattle worldwide [4]. In 1997, the zoonotic nature of C-type BSE, causing variant Creutzfeldt Jacob disease, was described [5].

The active BSE surveillance program in the European Union revealed two additional atypical BSE types in 2004, the so-called “H-type” in France [6] and an “L-type” BSE in Italy [7]. Pathological, molecular, and biological phenotypes of these unusual BSE types differ from C-type BSE, and in particular the molecular signature has been well-described and is used for TSE classification [6,7,8]. Atypical BSE occurs worldwide in older cattle (average age at detection 12.05 years) and numbers of detected animals seem to be unchanging, with 1–2 cases per million in tested bovines over eight years of age [9,10]. These epidemiological data led to the hypothesis that atypical BSE belongs to the group of spontaneously arising TSEs [9].

TSEs are characterized pathologically by degenerative changes in grey matter, including spongiform changes of neuropil, vacuoles within neuronal perikarya, and an astrocytic response. The detection of amyloid plaques is variable and depends on the TSE form [11]. While vacuolar lesion profiles in the brains of conventional mice inoculated with different TSE strains have been successfully used for strain typing purposes [5,12], lesion profiles in natural scrapie cases in sheep are highly variable and influenced not only by the host genotype but also by other yet-unknown factors [13,14,15]. Interestingly, for C-type BSE, several studies from different countries indicate a highly stable strain [16,17,18,19,20]. Unfortunately, almost all atypical BSE cases are fallen stock, therefore suitable material for performing a lesion profile is rather limited and the few results are insufficient for a comparative study [7,21,22,23]. On the other hand, the results of intracerebral studies are difficult to compare, because these studies frequently included different disease time points, at which the animals were sacrificed and different brain regions were used for analysis, in addition to being interpreted by various readers. Nevertheless, there is evidence of differences in the distribution of lesions between BSE types in specific brain regions, e.g., forebrain or cerebellum [24,25,26,27,28,29,30,31] (see literature review in Appendix A).

General characteristic for all TSEs is the massive accumulation of PrP^Sc^ in the central nervous system of clinically diseased animals. Over the years, differences in the neuroanatomical distribution, as well as the existence of several morphological PrP^Sc^ types, had been demonstrated by immunohistochemistry, the most variable in classical scrapie cases [32]. These patterns include cell-membrane/extracellular PrP^Sc^ types, i.e., neuropil-associated (linear, fine, or coarse particulate, coalescing, perineuronal), glial-cell associated (stellate, perivacuolar, subpial, subependymal, perivascular), ependymal-cell associated (supraependymal) and endothelium associated (vascular plaques) PrP^Sc^ depositions. Extracellular plaque-like accumulations of PrP^Sc^ have also been described. Furthermore, intracellular accumulation has also been reported with intraneuronal, intra-astrocytic, and intraglial accumulations [32,33]. These patterns are considered to be linked to cell tropism and PrP^Sc^ processing and had been useful in characterizing scrapie strains in the natural hosts [34,35,36]. In addition, transgenic mice and bank voles inoculated with TSE isolates are increasingly being subjected to PrP^Sc^ profiling to characterize TSE strains [37,38,39]. PrP^Sc^ profiles of different brain regions have already been described for all BSE types [7,8,19,22,23,24,25,26,27,28,29,30,31,40,41,42,43,44,45,46,47], but a systematic comparison between these studies is even more complicated than for the lesion profile because of the numerous variables (i.e., PrP^Sc^ types, available brain regions, grading, readers); moreover, the published results also contradict each other (see literature review in Appendix A). However, there are indications that some differences in the PrP^Sc^ profile, particularly in the cerebrum and cerebellum, may exist [7,24,29,30].

In total, 144 atypical BSE cases have been confirmed worldwide since 2001, and the yearly numbers of reported H- and L-type BSE cases have been stable over the years [10]. These statistics indicate that atypical BSE will not decline over the years, as was the case with C-type BSE. Atypical BSE is able to cross the species barrier to several experimentally challenged animals and an aetiological relationship between atypical and C-type BSE is still not finally clarified [9]. Therefore, in 2013, Regulation (EC) No. 999/2001 was amended, requiring that the BSE type be identified in BSE-confirmed samples in the European Union and elsewhere. This discriminatory testing relies solely on the molecular signature of the BSE types, which includes the determination of the protease-resistant size and the glycosylation pattern of PrP^Sc^ [6,7,8]. However, these techniques are not established in all National Reference Laboratories, and the availability of native reference material of atypical BSE is scarce and rapidly declining. Moreover, situations may arise where native material is not available (current or retrospective) or biochemical analysis yields unclear results. An additional tool to distinguish the case in question is then very helpful. Therefore, we report here a comparative systematic histopathological and immunohistochemical study of the brains of intracerebral C-, H-, and L-type BSE-infected cattle. For comparison, we included orally with C-type BSE infected animals. The results presented clearly indicate the discriminatory potential of tissue samples fixed in formalin, which will be useful to support future BSE classification efforts, in particular in cases where no frozen material is available.

## 2. Materials and Methods

All animals used in this study showed clinical signs of BSE and have been confirmed as BSE-positive by IHC and/or immunoblot. In Table 1, the most important details of the animal experiments are summarized.

### 2.1. Ethical Statement

All animal experiments described below have been performed in accordance with ethical principles and the animal welfare regulation. The experiments at the FLI in Germany took place before 2010, therefore the EU Treaty series 123 and, in particular, EU council 86/609/EEC for the protection of animals used for experiments and related national legislation were applied, which included a supervision by an Animal Welfare Officer. Moreover, the study was approved by the competent authority of the Federal State of Mecklenburg-Western Pomerania (LALLF M-V/TSD/7221.3-1.2-008/07 and LALLF M-V/TSD/7221.3-020/06).

For the experiments at the CFIA in Canada, all procedures involving experimental cattle were approved by the Canadian Food Inspection Agency Lethbridge Laboratory Animal Care Committee; all procedures adhered to the guidelines established by the Canadian Council of Animal Care.

### 2.2. Intracerebral BSE Challenge

FLI/Germany: female Holstein-Friesian calves six months of age were inoculated intracerebrally with 1 mL of a 10% brain homogenate derived from confirmed atypical BSE cases were used for this study. A total of five animals with L-type BSE (R172/02) and five calves with H-type BSE (R152/04) were inoculated. Both original cases have already been described in detail [8], as has the experiment [48]. Shortly, the intracranial inoculation was done under ketamine/xylazine anaesthesia with support by physiological saline solution administered intravenously throughout the operation. The forehead was shaved and disinfected prior to surgery and the optimal location of the injection site was calculated as described before [49,50]. The cerebral cavity was accessed next to the bone septum by the shortest route through the frontal sinus, targeting the rostral midbrain. A dental drill was used to drill a hole of 1 mm in diameter through both bone layers of the skull. A syringe with a 9 cm needle was used to inject the brain homogenate into the rostral midbrain, while slowly retracting the needle. After the inoculation, the skin was sutured with stitches and covered with a Band-Aid. During the incubation time the animals were closely monitored for the development of neurological signs associated with BSE, and when the ethical termination criteria were met, the animals were euthanized and necropsied under TSE sterile conditions.

CFIA/Canada: Five Hereford x Angus weaned calves of approximately 2–3 months of age and six Hereford x Angus weaned calves of approximately 5–6 months of age were obtained from the CFIA Lethbridge SPF herd for two separate BSE animal challenge experiments in 2005 and 2009, respectively. Calves were tagged and visually inspected upon arrival and moved into the BSL-3 containment facility at the CFIA Lethbridge Laboratory. All animals were kept in a single group to minimise stress. Animals were given water and feed ad libitum and were kept for 7 days to acclimatise to their new surroundings. Animals were observed daily starting from the acclimatisation period and throughout the project. Any abnormalities in behaviour were reported to the supervising veterinarian. Once acclimatised, the calves were divided into groups for the different BSE inoculates. The calves were intra-cranially challenged with 1 mL of a 10% (*w*/*v*) brain homogenate of either C, H, or L-type BSE, each derived from one BSE confirmed cases. Calves were sedated with a ketamine and xylazine mixture in the animal room and the surgical site was prepared as per standard protocols. A local anaesthetic (Lidocaine) was injected into the skin around the incision site. A midline incision was made in the skin at the junction of the parietal and frontal bones and a 1.6 mm diameter hole was drilled through the calvarium to gain access to the brain. The inoculum was injected into the midbrain using a 22-gauge, 3.5-inch long needle, while slowly retracting the needle from the brain. The skin incision was closed with a single suture and antibiotic treatment (topical and systemic) was provided. Animals were allowed to recover in the animal room under observation. Animals were held in the BSL-3 containment room for 30 days following inoculation, at which time they were moved to a secure outdoor facility. Animals in each project were kept in a single pen, surrounded by an 8 foot secured perimeter fence. Calves were observed daily for the development of clinical signs associated with BSE and maintained in the facility until the established termination criteria were met. The animals were then euthanized and examined post-mortem.

### 2.3. Oral Animal BSE Challenge (FLI)

Three clinically affected Simmental cross-breed cattle were included in this study to compare if the identified brain lesions and PrP^Sc^ profiles differed between both inoculation routes. Details of this study have been published earlier [51], and the most important anamnestic data are summarized in Table 1.

### 2.4. Tissue Selection

To obtain indications of which brain regions are most promising for discriminative studies, previously published lesion profiles and PrP^Sc^ profiles of all three BSE types were reviewed and summarized in Appendix A. The brainstem, being the sample routinely analysed during active surveillance, was included to provide a reference to previous studies. We also included cerebrum and cerebellum based on the literature review, as well as red nucleus, hippocampus, and septal nucleus, which were included due to previous observations in field and experimental cases at the CFIA and FLI. In total, six brain regions with 17 neuroanatomical localisations of interest were included in the study (Table 2, Appendix A). These regions consistently had sufficient amounts of PrP^Sc^ and were promising for BSE classification by immunohistochemistry. For some regions (septal nucleus at the CFIA, hippocampus at the FLI) paraffin blocks were newly prepared from brain tissue fixed in 10% neutral buffered formalin, but most samples were chosen from archived paraffin blocks at the CFIA and the FLI. If a particular brain area was not available, this was documented. After decontamination with 98% formic acid followed by running tap water for at least 40 min, formalin-fixed samples were dehydrated before being embedded in paraffin. Further processing of all wax blocks was done at the CFIA and a set of successive 3 µm sections were prepared from each block.

### 2.5. Histopathological and Immunohistochemical Analysis

A total of five readers examined all brain regions semiquantitatively for the total amount and the cellular pattern of PrP^Sc^ deposition. All results were discussed among the readers, and conflicting results were collectively clarified. However, to avoid discrepancies based on individual quantitative classifications, all graphs presented are based on the results of one representative reader.

#### 2.5.1. Histopathology/Lesion Profile

One section of each block was stained with Hematoxylin and Eosin for histopathological analysis. For lesion profiling, the extent of lesions in each brain region and for each animal was determined using the following score: weak (score 1.0), mild (2.0), moderate (3.0), and severe (4.0). Based on the scores obtained, an average score for each brain region was calculated for all animals in the respective group (C-, H-, and L-type, and orally infected C-type, respectively) for the graphs.

#### 2.5.2. Immunohistochemistry/PrP^Sc^-Profile

For each block a highly sensitive Immunohistochemistry was applied using the monoclonal anti-PrP antibody F99/97.6.1 (VMRD Inc, Pullman, WA, USA), which recognizes an epitope at the core region of the protein. During the pre-treatment process, formalin-fixed paraffin-embedded tissue sections on glass slides were autoclaved at 121 °C for 25 min in citrate buffer (pH 6.0). F99/97.6.1 was applied at a dilution of 1:18,000 TBST (Tris buffered saline including Tween). A highly sensitive detection system using an HRP-linked secondary antibody (EnVision HRP; Dako Diagnostics) was used; slides were developed using 3, 3′-diaminobenzidine tetrahydrochloride (DAB) as the chromogen and counterstained with Gill II haematoxylin. All sections were examined by light microscopy.

Based on the PrP^Sc^-types described previously [32], eleven different reaction patterns were identified and included in the study. These are intraneuronal (ITNR), intra-astrocytic (ITAS), intramicroglial (ITMG), stellate (STEL), submeningeal (SMENG), fine particulate (f-PART), coarse particulate (c-PART), coalescing (COL), perineuronal (PNER), linear (LIN), and plaque-like (PL). Appendix A shows examples of the reaction patterns examined. The extent of different cellular PrP^Sc^ patterns in each brain region was determined for each animal, and the following scores were used: weak (score 1.0), mild (2.0), moderate (3.0), and severe (4.0). These scores were then totalled to determine the total amount of PrP^Sc^ accumulation for each brain region for each animal. For the graphs, the average of all animals per group (C-, H-, and L-type, and orally infected C-type, respectively) was determined for each brain region for both cellular pattern and total amount of PrP^Sc^ deposits.

## 3. Results

### 3.1. Lesion Profile

The vacuolation profiles of the three BSE types were very similar to each other. We only observed a clear difference between classical and atypical BSE-infected groups in the molecular layer of the cerebrum (frontal cortex) and in the septal area with distinct lesions in H- and L-type and no or only weak involvement in C-type BSE (Appendix A).

We also compared the lesion profile determined for animals challenged orally and intracerebrally with classical BSE (Appendix A). Whereas brainstem, cerebellum, frontal cortex, and septal area showed only minor differences in vacuolar pattern, in orally infected cattle the red *nucleus* was distinctly more involved, whereas *Hilus* and *Str. oriens* were clearly less involved.

### 3.2. PrP^Sc^-Profile/General Remarks

All animals showed clear signs of PrP^Sc^ accumulation in all regions examined, ranging from weak (*n* = 1), mild (*n* = 3), and moderate (*n* = 7) up to severe (n = 10). For more details see Table 1.

The distribution of the PrP^Sc^ accumulation throughout the brain was very similar for the three BSE types upon intracerebrally challenge, with slightly lower amounts of PrP^Sc^ levels observed for L-type BSE in the red nucleus (Appendix A).

Comparing the overall distribution of PrP^Sc^ depositions in the orally and intracerebrally C-type BSE inoculated cattle, a slightly lower magnitude of PrP^Sc^ accumulation was seen in the hippocampal parts upon being orally challenged. All other regions investigated showed very similar results (Appendix A).

Nine of the 17 brain regions studied showed differences in PrP^Sc^ profiles between BSE types that could be used for discriminatory purposes and are presented here in detail. Table 3 gives an overview of the most important PrP^Sc^ profile characteristics potentially useful for differentiation.

The most characteristic pattern of C-type BSE is the higher variability of staining reactions as compared to atypical BSE cases. In several brain regions C-type animals showed a STEL reaction pattern which is only rarely seen in H-type cases and is absent in L-types. Additionally, C-type animals frequently show, mostly associated with high PrP^Sc^ levels, a coarse PART PrP^Sc^ deposition often accompanied by distinct LIN staining reactions and occasionally even PL pattern.Cattle infected orally with C-type BSE show little difference in the PrP^Sc^ profile as compared to intracerebrally inoculated animals. This particularly concerns the PART staining reactions which, in some regions (hypoglossal nucleus, molecular layer of cerebellum, septal area), also show a coarse deposition in intracerebrally infected animals, whereas in orally infected animals only fine PART distributions are seen. Additionally, in the hilus of hippocampus and white matter of frontal cortex, a STEL pattern was only visible in intracerebrally inoculated animals.Animals infected with L-type BSE show a fine and equally distributed PART PrP^Sc^ accumulation in several regions which always, even in advanced cases, i.e., in moderately to severely affected animals, has a regular appearance. LIN staining reactions are frequently seen, but not as distinct or typical as in C-type BSE cases. In contrast, PL are rarely seen and confined to certain regions (external grey matter and white matter of cerebrum).The most typical pattern of H-type animals is an extensive ITMG staining reaction most often in combination with a fine PART neuropil accumulation of PrP^Sc^, which allows the reader to recognize this pattern even in low magnification. LIN staining reactions are rarely seen, but PL patterns are common, often in the form of very small plaque-like accumulation.

### 3.3. Hypoglossal Nucleus 

PrP^Sc^ accumulations were seen in all animals ranging from mild to severe, but L-type infected animals always showed a minor involvement of this nucleus in contrast to H- and C-type challenged animals (Figure 1A1–A4). However, the most striking differences between the BSE types concern the ITNR staining reaction, which was completely absent in L-type animals, but could affect, depending on the degree of the overall infection, almost all neurons in C- and H-type cases. Additionally, in H-type animals, a pronounced severe ITMG staining reaction in combination with a distinctly lower and fine PART accumulation was obvious. This pattern is in contrast to C- and L-type BSE cases, which showed mild to moderate IMTG reactions embedded in comparable PART accumulation. It is worth mentioning the mostly fine PART deposition pattern in atypical cases, whereas C-type infected animals also have, depending on the degree of accumulation, a coarse PART reaction pattern. In this regard, C-type animals revealed the most variable pattern, and STEL and LIN were also detectable. However, these differences are only seen in moderately to severely affected cases.

### 3.4. Cerebellum 

The cerebellum can be used to discriminate between the BSE-types and, in particular the molecular and granular layers, which showed a characteristic pattern type for every BSE. However, these characteristics can clearly be seen only in advanced (from mild upwards) cases. Weakly affected animals showed a clear PrP^Sc^ accumulation in the deep cerebellar nuclei only and, if any at all, singular, multifocal distributed PrP^Sc^ depositions in all other neuroanatomical sites. In our study one animal did not show any PrP^Sc^ deposition in the molecular layer, and two animals were negative in the granular layer. Several animals displayed a weak PrP^Sc^ accumulation (seven animals in the molecular layer and four in the granular layer), which were confined to single cells and/or localizations. However, the majority of cases presented mild to severe amounts of PrP^Sc^, which were detectable throughout the cerebellum.

In the *molecular layer* (Figure 1B1–B4), the most characteristic feature in C-type infected animals was the variability in the reaction patterns. A distinct STEL staining was detected in combination with an ITMG deposition, a fine to coarse multifocal PART, and a randomly distributed LIN staining reaction. In contrast, a STEL reaction pattern was not seen in L-type animals. Instead, the characteristic pattern included a fine homogenous PART reaction pattern plus some fine LIN staining. With one exception (see below), H-type animals revealed a mild ITMG reaction pattern close to the granular layer and mostly combined with randomly distributed fine, rarely coarse, PART accumulations.In the *granular layer*, the most obvious difference was the PNER reaction pattern in L-type infected animals, in which an ITNR PrP^Sc^ deposition was rarely seen. This is quite the opposite to C-type BSE, which induced a distinct ITNR, but never a PNER staining reaction. In H-type cases, both variants were seen. In all BSE cases, granular cells showed intracellular PrP^Sc^ accumulation, which was most distinct in H-type cases. Only C-type-affected animals displayed a widespread net-like staining reaction following the dendrites of the granular cells and resembling a STEL reaction pattern. L-type BSE induced a fine, homogeneously distributed PART PrP^Sc^ accumulation that involved almost the entire layer in advanced cases, whereas C- and H-type animals showed fine to coarse PART depositions that were randomly distributed. Additionally, PL patterns were seen in C- and H-types only but were most prominent in C-types.In atypical BSE cases, an ITMG staining reaction was widely distributed in the *white matter of the cerebellum* and involved in moderate to severe cases almost all glial cells. This is in clear contrast to C-type BSE, in which this region is never more than mildly affected. More distinctly, a PNER staining reaction was only seen in L-type cases.In *cerebellar nuclei* and adjacent white matter (Figure 2A1–A4), a high variability of staining reactions was seen with C- and L-type BSE. In contrast, H-type BSE revealed a prominent ITMG PrP^Sc^ deposition in combination with a fine PART PrP^Sc^ background.

One clinical H-type infected animal (29033) showed clear deviations from the described pattern with additional STEL and LIN depositions in the molecular layer and a fine homogenous PART staining reaction of the granular layer, mixed with a multifocal PNER reaction. Moreover, PL was seen in both layers.

### 3.5. Red Nucleus 

In two animals, the red nucleus was not available. When present, distinct amounts of PrP^Sc^ were detected (Figure 2B1–B4), especially in C- and H-type, ranging from mild to severe. However, in L-type BSE, this area was always less affected.

A distinct pattern supporting discrimination is the pronounced ITMG staining reaction in H-type BSE cases, which is usually embedded in a fine PART background. An ITMG PrP^Sc^ accumulation is also visible in C-type infected animals, but this reaction pattern is accompanied by an extensive coarse PART background and a distinct LIN reaction.

### 3.6. Hippocampus 

In one animal the *hilus* area was not available, and two additional animals did not show any PrP^Sc^ deposition. All other cases showed mild to severe accumulations of PrP^Sc^, affecting the hippocampus entirely (Figure 3A1–A4).

The most affected area in all BSE-types was the *Str. radiatum*, and the lowest affected was the *Str. oriens*. The neuroanatomical distribution of PrP^Sc^ in the hippocampus showed clear differences between the BSE types. While *Str. radiatum* is affected in all BSE types, atypical BSE cases have an additional preference to *Str. pyramidale* and C-type BSE to the hilus (dentate polymorphic layer). Moreover, C-type associated PrP^Sc^ accumulation in the hilus was mainly confined to the central parts of this area, whereas in atypical cases PrP^Sc^ is more likely to be found in the peripheral zone. These differences can even be seen in mildly infected animals. The cellular PrP^Sc^ pattern also revealed distinct differences between the BSE types. STEL, multifocally distributed in all neuroanatomical structures, as well as PL in the hilus, were only seen in C-type infected animals. In contrast, the most pronounced staining reaction for H-type was ITMG, which was evenly distributed in the entire hippocampus. Fine PART staining reactions were characteristic for atypical cases, whereas in C-type BSE, coarse accumulations were also detectable.

### 3.7. Septal Area 

The septal area was not always well preserved in all animals, which included a suboptimal size (*n* = 4) or an ambiguous identification (*n* = 4). In one animal, this area was not available at all. Mild to moderate PrP^Sc^ accumulations were detectable in the remaining animals (Figure 3B1–B4).

All BSE types showed fine/coarse PART, LIN, and ITMG staining reactions. However, only the C-type BSE infected animals revealed STEL PrP^Sc^ depositions. A characteristic for H-type BSE is the distinct ITMG staining, involving almost 100% of the glial cells. This is in clear contrast to L-type BSE, which mostly induced a fine PART PrP^Sc^ deposition and mild amounts of ITMG staining reactions.

### 3.8. Frontal Cortex 

One animal lacked completely detectable PrP^Sc^ in the *molecular layer*, and three animals revealed very low amounts that were confined to randomly scattered cells/localizations. In all other cases, mild to moderate PrP^Sc^ accumulations were found to affect the entire section, but multifocal spots with significantly higher amounts of PrP^Sc^ were also seen. 

A PrP^Sc^ “tape-like pattern” in the most external parts of the *molecular layer*, most likely associated with horizontal projections of Cajal cells, was characteristic of C- and H-type BSE infected animals. This “tape” consisted of fine to coarse PART, as well as ITMG deposits. In contrast, in L-type BSE a mostly fine PART staining reaction in combination with a ITMG accumulation was homogeneously covering all parts of the layer. An ITNR staining reaction was seen with all types, but it was seen in L-type BSE to a higher degree, even in very mild cases. A STEL reaction pattern was clearly seen with C-type infected animals. A characteristic for H-type BSE was the detection of very small but prominent and homogeneously distributed PL staining reactions. In C- and L-types this reaction pattern was seen to a much lesser degree and was confined to single locations. In Figure 4A1–A4 representative cases from each BSE type are presented.

## 4. Discussion

Over the last decades, several studies have been performed that describe the histopathological and immunohistochemical characteristics of classical and atypical BSE. In C-type BSE, sufficient amounts of formalin-fixed and paraffin-embedded material from almost all brain regions are available, from field cases and orally and intracerebrally-challenged animals. However, in atypical BSE field cases, suitable material for such studies are not only very scarce, but most published results had to be limited to the brainstem if suitable material was available at all [7,21,22,23]. On the other hand, studies using material from intracerebral inoculation often lack a common systematic approach because they often focused on different brain regions for read-out, different disease time points when animals were sacrificed, or different classification and profiling systems [24,25,27,28,29,30,31]. In addition, an intracerebral inoculation is an artificial experimental setting, and it is not known whether brain samples from such studies are comparable to BSE in field cases at all. Therefore, filling these knowledge gaps about atypical BSE in pathology was the aim of our study. Using a systematic and comprehensive approach, we were able to demonstrate not only that C-, H-, and L-type BSE can be distinguished by immunohistochemistry, but our data also show that intracerebral BSE infection most likely reflects the natural disease quite well in terms of both lesion and PrP^Sc^ profile. However, it should be kept in mind that our study is based on very well-prepared (i.e., non-autolytic) material obtained by closely supervised animal experiments. Therefore, our results still need to be verified using field cases that are much more diverse in terms of the timing of the disease at the time of killing and the state of preservation of the brain. Nevertheless, the results presented here can be used as reference data to help pathologists in discriminating atypical and classical BSE cases in animals where native material is not available (actual or retrospective cases), in laboratories in which the biochemical method is not established, and/or in cases with ambiguous results obtained by biochemical analysis.

For C-type BSE, the previously reported lesion profiles were very congruent, independently of researchers and origin and/or breed of cattle [16,17,18,19,20]. Five (hypoglossal nucleus, nucleus of the solitary tract, nucleus of the spinal tract of the trigeminal nerve, frontal cortex, and septal area) of the 17 brain structures examined in our study were also characterized in those previous reports and showed similar lesion profiles in both intracerebrally and orally infected classical BSE (data not shown). The most obvious difference between intracerebrally and orally infected animals in the extended lesions profiles applied in our study is confined to the red nucleus, which showed very prominent vacuolation in orally infected animals, which was much less pronounced in the intracerebrally infected groups. In this regard, it should be remembered that red nucleus neurons physiologically develop intracytoplasmic vacuolization with age ([52], personal observation) and that orally infected animals are almost twice as old as intracerebrally infected animals because of the longer incubation time. The age-related changes in these animals may be superimposed and amplified by the disease-related alterations, explaining the pattern observed here. The fact that the intensity of PrP^Sc^ accumulation in this region is very similar for both inoculation routes also support this theory. In addition, parts of the hippocampus in orally infected animals had distinctly fewer lesions than in intracerebrally infected animals, which is also reflected in the overall lower distribution of PrP^Sc^ here, which normally precedes the development of lesions ([53], personal observation). Such distribution dynamics are also supported by previous observations of C-type BSE field cases describing rostral parts of the brain, including the cerebrum, which was less affected as compared to brainstem and midbrain [11,16,18,19]. Moreover, for sheep infected intracerebrally with scrapie or BSE, it has been described that the initial target sites of PrP^Sc^ accumulation in the brain are the same, regardless of the route of inoculation, but an overall greater amount of PrP^Sc^ accumulation, particularly in forebrain areas, was observed in intracerebrally infected animals as compared to orally infected animals [32]. In this regard, it has to be kept in mind that after oral ingestion of PrP^Sc^, the accumulation process in the central nervous system starts with single affected neurons in the brain stem [51,54]. However, because the hippocampus is generally only indirectly associated with these earliest sites of PrP^Sc^ accumulation, it is reasonable to speculate that PrP^Sc^ has taken longer to spread to this brain region than to the neuroanatomical structures directly associated with the first affected cells. In intracerebrally infected animals, on the other hand, PrP^Sc^ is administered comparatively broadly, most likely affecting multiple brain regions simultaneously since the beginning. Thus, a subsequent faster and broader distribution of lesions and PrP^Sc^ in clinical stages of disease, as shown in our study, seems logical. This would also explain the presence of a more pronounced coarse particulate reaction pattern in intracerebral infected animals, a pattern which is usually seen in more advanced cases. Taken together, as shown by others in different experimental set-ups [28,32,55,56,57,58], the two different routes of inoculation used here did not have a major impact on the preferences of PrP^Sc^ accumulation and subsequent lesions in C-type BSE, and only the magnitude and overall broader distribution seems to be more pronounced in intracerebral infected cattle.

More surprising was the fact that intracerebrally infected classical and atypical BSE cases showed large similarities not only in the lesion profiles, but even more so in the overall distribution pattern of PrP^Sc^. Despite all the differences in the molecular signature of the BSE types, the final preferences in terms of brain target sites seem to be the same. This is in marked contrast to classical and atypical scrapie in sheep [13,36,59,60] and may be due, at least in part, to the absence of polymorphisms in the bovine prion protein gene, which are known to influence the pathological phenotype [32]. Slight differences were only seen in the molecular layer of frontal cortex and septal area, in which atypical BSE induced more pronounced lesions as compared to C-type BSE. However, this pattern is not reflected by the magnitude of PrP^Sc^ deposition, which shows a moderate amount of PrP^Sc^ for all types in these structures. The affection of distinct cortical lesions in H- and L-type BSE were also described by others using the same inoculation scheme [25,26,27,29], and Konold and colleagues [30,31] even pointed out that this pattern was more distinct as compared to C-type BSE-infected animals inoculated by the same route. In contrast, field data [7] and results from an orally L-type challenged animal [47] showed only mild lesion in the frontal cortex, but both animals were overall only mildly affected. These results indicate that certain brain regions are more susceptible to atypical BSE-induced lesions than to classical BSE.

Numerous studies have described the immunohistochemical PrP^Sc^ profile of C-, H-, and L-type BSE cases. However, a systematic comparison of these results is nearly impossible due to numerous variables and the lack of a common reference, and often leads to contradictory results (see the literature review in the supplement to this publication). To ensure the utility of our study we therefore defined, based on personal experience, the PrP^Sc^ phenotypes and the neuroanatomical sites to be included, as well as the method of quantification, before starting the assessment. In doing so, we detected that the overall magnitude and distribution of PrP^Sc^ revealed a remarkable consistency between the BSE types, and therefore cannot be used for differentiation. However, the PrP^Sc^ profile, which focuses on the cellular reaction pattern within the different neuroanatomical structures, revealed distinct differences between the BSE types.

H-type BSE induced in nearly all neuroanatomical sites investigated a prominent intramicroglial reaction pattern, which was, in most cases, combined with a fine particular PrP^Sc^ deposition. The resulting pattern is very characteristic and easily recognizable, even in low magnifications. An extensive intramicroglial and stellate reaction pattern was already described by others [26,31,61], but for the neuropil PrP^Sc^ depositions, the descriptions in literature range from not further specified particular reactions [27,30] or aggregates [8] to fine [22], fine to coarse [45], and coalescing [27]. All mentioned results are based on C-terminal antibodies, and it remains unclear whether the differences are due to different definitions of the particulate PrP^Sc^ deposition profile and/or subjective assessments of different readers. Multifocally, we also detected mild, coarse, and even small plaque-like patterns depending on the neuroanatomical location, but in the overall view, the combination of intramicroglial and fine particulate PrP^Sc^ deposition predominated, which only gave the aforementioned characteristic picture when considered together. However, it has to be kept in mind that we still do not know much about potential H-type BSE strains, and these differing results may also be an indication that the different H-types studied may elicit different reaction patterns. In this regard, H-type infected animal 29,033 in our study is of particular interest, as it showed some similarities with the pattern usually seen in C-type infected cattle. A systematic investigation including H-type BSE cases from all over the world and using the same systematic approach could clarify that question.

Detailed descriptions of L-type BSE profiles are scarce in literature, and several authors mostly refer to the detection [24,25,27,29,30] or failed detection [46,47] of plaques or plaque-like structures, which were prominent in the very first L-type BSE case [7]. It has to be emphasized that in our study, plaque-like formations induced by L-type BSE were characteristic only in the internal grey layer, as well as the white matter of the cerebrum, and that they were also much larger and occurred much more frequently than in the other BSE types. This restricted focal distribution of plaques/plaque-like structures was also mentioned by others in intracerebral infected animals [24,27,30], but a multifocal plaque formation in different parts of the brain was also described in field cases [7] and after intracerebral infection [29]. Furthermore, all variants of a particulate neuropil PrP^Sc^ accumulations were described previously [25,29,30,31,42,47]. However, we predominantly observed a fine particulate, often even regularly distributed PrP^Sc^ accumulation in L-type BSE affected cattle, frequently together with fine linear staining reactions and a mild intramicroglial pattern. In contrast to H-type BSE, this pattern is specifically seen in certain neuroanatomical sites, namely in the hypoglossal nucleus, molecular and granular layers of the cerebellum, and the septal area. In these structures, the proposed similarity of the PrP^Sc^ profile to C-type animals [46] cannot be seen. Additionally, worth mentioning is the intraneuronal and perineuronal tropism, which is particularly conspicuous in certain neuroanatomical structures of L-type BSE. For example, in the hypoglossal nucleus, C-terminal antibodies failed to detect any intraneuronal PrP^Sc^ depositions, but this pattern can easily be found in C- and H-type animals. Perineuronal staining reactions, which have also been described by others [24,31,47], occur most strikingly and clearly in neuronal cells of the cerebellar granular layer, and only in L-type infected cattle. As described for H-type infected animals, it is reasonable to associate the observed variations to previous reports with different definitions of cellular PrP^Sc^ accumulations and with the lack of comparability between the different studies. However, it should be kept in mind that variations in the underlying L-type cannot be excluded here either.

Numerous publications are available describing the PrP^Sc^ profile of classical BSE from field cases [19,25,26,40,41,42,43,44], but descriptions of intracerebrally infected animals are rare [24,28]. Besides different variants of particular accumulation (fine to coarse, coarse, coalescing), a mixture of linear, stellate, intraglial, intra- and perineuronal staining reactions have been described for all examined structures. This is in close agreement with our observations, in which C-BSE induced the most variable PrP^Sc^ profile of all BSE types throughout the brain, independently of the inoculation route. However, the most characteristic pattern in our hands for C-type BSE is the widely distributed detection of stellate PrP^Sc^ depositions. In literature, a perineuronal staining reaction is frequently mentioned [19,26,40,41,43,44], but this pattern is mostly described for neuroanatomical sites not included (thalamus, hypothalamus, nucleus caudatus-putamen) in our study or not further detailed (DMNV, nucleus solitary tract, reticular formation) here. Neuroanatomical sites with distinct perineuronal C-type PrP^Sc^ depositions in our study were the red nucleus and the internal layer of the cerebrum. However, at these sites, atypical BSE induced the same pattern. Plaque and/or plaque-like PrP^Sc^ depositions were also described in the midbrain [25,40], as well as in the thalamus and cerebrum [40]. Accordingly, we saw mild plaque-like staining reactions in several brain regions, but in the hippocampus (hilus) this pattern was most pronounced and was unique in animals infected with C-type BSE independently of the inoculation route. In summary, the results presented here for C-type infected animals, which are highly in agreement with previous reports, again underline the high consistency and stability of C-type BSE.

In addition to the cellular PrP^Sc^ profile, the topographic distribution in specific neuroanatomical regions also deserves attention with respect to its differentiation potential therein. For example, in advanced cases, the molecular and the granular layer of cerebellum showed a very fine homogeneous diffuse PrP^Sc^ depositions with L-type BSE, including a unique randomly scattered perineuronal PrP^Sc^ deposition in the granular layer, which is in clear contrast to the multifocal variable staining reaction seen with C-type, and the strong intramicroglial reaction pattern induced by H-type BSE. Similar differences were also seen by others [30], and an inclusion of the analysis of this region in active surveillance programs may therefore be helpful for differentiation purposes. In the molecular layer of the cerebral cortex the PrP^Sc^ deposition seems to be associated with the horizontal projections of Cajal cells forming a tape-like pattern in C- and H-type BSE infected animals. Moreover, the distribution pattern is very characteristic within the hippocampus, in particular the dentate polymorph layer, with C-type PrP^Sc^ accumulating in the central part of this site, whereas atypical BSE is confined to cells in the periphery. To the best of our knowledge, these patterns have not yet been described.

The different PrP^Sc^ profiles detectable in different TSE strains at specific neuroanatomical sites or in specific cells have previously been attributed to differences in cell tropism and processing of the PrP^Sc^ present, and are thought to result from conformational variabilities of the underlying PrP^Sc^ [15,32]. The three BSE types studied here showed, in comparison to classical scrapie [32], a very limited cellular PrP^Sc^ profile, which is in agreement with previous studies [19,24,25,26,28,40,41,42,43,44]. Interestingly, all BSE-types induce PrP^Sc^ accumulations in the same neuroanatomical target sites, most probably due to the distribution of PrP^C^ at these sites and the general functional ability of all BSE types to infect both neurons and glial cells or to deposit in the neuropil. Nevertheless, a close and systematic examination clearly revealed the qualitative differences in cell tropism associated with specific neuroanatomical localizations, e.g., the absence of an intraneuronal staining reaction in the hypoglossal nucleus infected with L-type BSE or the restriction to a specific subset of neurons in the hippocampus in atypical BSE cases, which are consistent with the aforementioned theory [15,32]. It appears that the three types of BSE favour specific subsets of neurons and/or glial cells (or vice versa). Of particular interest in this context is the strong ITMG PrP^Sc^ deposition observed in H-type BSE with minimal (fine PART) extracellular deposition in the neuropil. This may suggest that uptake of PrP^Sc^ by glial cells is facilitated in this type of BSE, preventing major deposition in the neuropil (characterized by coarse or coalescing PART reaction patterns). However, the extent to which this is related to differences in the PK cleavage site known from biochemical analysis cannot be answered here.

## 5. Conclusions

Our systematic examination of the PrP^Sc^ profiles of the C-, H-, and L-type BSE showed clear qualitative differences in a selection of neuroanatomical structures, which includes all parts of the brain and can easily be used for an additional discriminatory purpose. In brief:C-type BSE showed the most variable PrP^Sc^ profile with a distinct tendency for a stellate pattern, which is particularly obvious in the multifocal distribution pattern found in the cerebellar cortex. In addition, hypoglossal nucleus revealed clear intraneuronal PrP^Sc^ accumulation, the central parts of the hippocampal hilus were massively affected even with unique plaque-like formations, and in the molecular layer of the cerebrum, PrP^Sc^ accumulations seemed to be associated with Cajal horizontal cells, forming a tape-like pattern.H-type BSE showed a characteristic strong intramicroglial PrP^Sc^ accumulation pattern throughout the brain, which is mostly associated with a fine particular staining reaction of the neuropil. In the hypoglossal nucleus an intraneuronal staining reaction was obvious, while in the hilus of the hippocampus, PrP^Sc^ accumulation was mostly confined to peripheral neuronal cells. Moreover, in the molecular layer of cerebrum, a PrP^Sc^ tape-like pattern is evident.L-type BSE revealed a mostly fine particulate and diffuse PrP^Sc^ accumulation in the neuropil in the molecular layers of the cerebellum and cerebrum. In the cerebrum, large plaque-like formations were frequently found, and in the granular layer of cerebellum, a unique perineuronal staining reaction was visible, while an intraneuronal PrP^Sc^ deposition in hypoglossal nucleus was missing.

It has to be kept in mind that most BSE cases are diagnosed in fallen stock animals, and whole brains will only be available in exceptional cases for further studies. Further studies should therefore focus on the liability of the discrimination scheme presented, using available samples from field cases. In any case, however, considering our results and in agreement with previous studies [30], we highly recommend including not only brainstem but also cerebellum in routine surveillance sampling. As demonstrated by the long-standing sampling procedure for the detection of TSE in small ruminants, this can be readily done through the Foramen magnum and could be used as an additional diagnostic tool to support the differentiation of BSE types.

## Figures and Tables

**Figure 1 pathogens-12-00353-f001:**
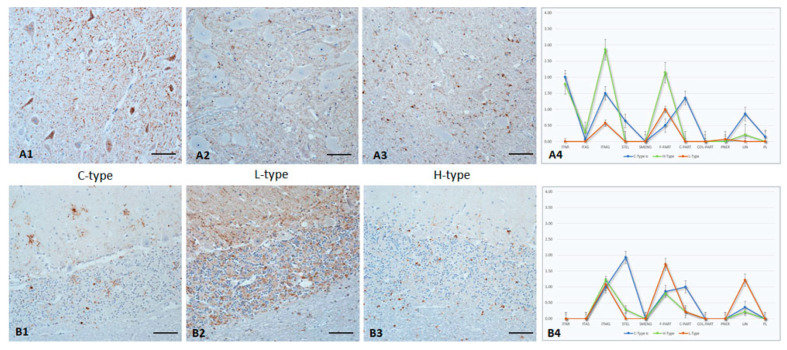
PrP^Sc^ accumulation in brainstem (**A1**–**A4**) and cerebellum (**B1**–**B4**) of C-, L-, and H-type infected cattle (Immunohistochemistry, mab anti-PrP F99). **A1**–**A3**: Brainstem/hypoglossal nucleus with a lack of intraneuronal staining reaction in L-type BSE, which is in clear contrast to C- and H-type, Bar 50 µm; **A4**: PrP^Sc^-profiles in hypoglossal nucleus of all three BSE types; **B1**–**B3**: Cerebellum/molecular layer with a diffuse PrP^Sc^ accumulation in L-type BSE, whereas C-type BSE shows a distinct multifocal and, in particular, STEL reaction pattern, and H-type BSE shows a prominent intraglial deposition. A similar pattern can also be seen in the granular layer, Bar 50 µm; **B4**: PrP^Sc^-profiles of cerebellum/molecular layer of all three BSE types. The extent of different cellular PrP^Sc^ patterns in each brain region was determined for each animal, and the following scores were used: weak (score 1.0), mild (2.0), moderate (3.0), and severe (4.0). For the graphs, the average value of all animals for each group (C-, H-, and L-type respectively) was determined for each brain region.

**Figure 2 pathogens-12-00353-f002:**
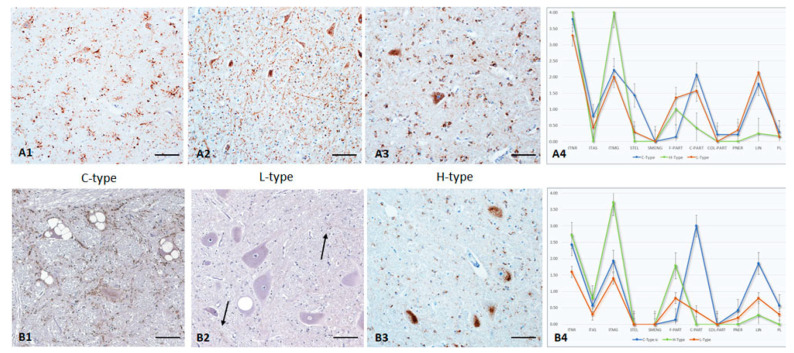
PrP^Sc^ accumulation in Cerebellar Nuclei and adjacent white matter (**A1**–**A4**) and Red Nucleus (**B1**–**B4**) of C-, L-, and H-type infected cattle (Immunohistochemistry, mab anti-PrP F99). **A1**–**A3***:* High variability of PrP^Sc^ deposition pattern in cerebellar nuclei and adjacent white matter in C- and L-type BSE, which is in clear contrast to H-type BSE with a distinct ITMG staining reaction and a background of fine-PART, Bar 50 µm; **A4**: PrP^Sc^-profiles of all three BSE types in cerebellar nuclei; **B1**–**B3***:* Red nucleus revealing a highly variable staining reaction in C-type BSE (oral infected cow, note the massive neuronal degeneration), in contrast to a multifocal fine PART PrP^Sc^ deposition in L-type BSE (arrows) and the prominent ITMG accumulation in H-type BSE, Bar 50 µm; **B4**: PrP^Sc^-profiles of all three BSE types in red nucleus. The extent of different cellular PrP^Sc^ patterns in each brain region was determined for each animal, and the following scores were used: weak (score 1.0), mild (2.0), moderate (3.0), and severe (4.0). For the graphs, an average value of all animals in each group (C-, H-, and L-type respectively) was determined for each brain region.

**Figure 3 pathogens-12-00353-f003:**
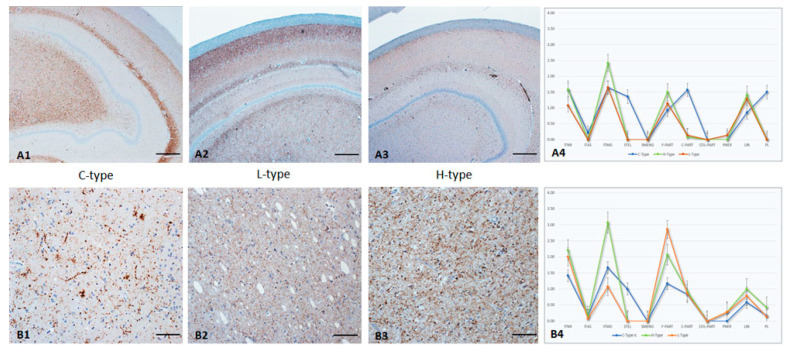
PrP^Sc^ accumulation in hippocampus/hilus (**A1**–**A4**) and septal nucleus (**B1**–**B4**) of C-, L-, and H-type infected cattle (Immunohistochemistry, mab anti-PrP F99). **A1**–**A3**: Hippocampus/hilus in C-type infected animals with a severe PrP^Sc^ deposition in more central parts, whereas L- and, in particular, H-type BSE induced a PrP^Sc^ accumulation in peripheral parts of the structure, Bar 100 µm; **A4**: PrP^Sc^-profiles in hippocampal hilus of all three BSE types; **B1**–**B3**: Septal area with a variable staining reaction induced by C-type BSE (LIN, STEL, coarse-PART), whereas atypical cases predominantly revealed a fine diffuse PART pattern in H-type, with a distinct intramigroglial component, Bar 50 µm; **B4**: PrP^Sc^-profiles of all three BSE types in septal nucleus. The extent of different cellular PrP^Sc^ patterns in each brain region was determined for each animal, and the following scores were used: weak (score 1.0), mild (2.0), moderate (3.0), and severe (4.0). For the graphs, an average value of all animals in each group (C-, H-, and L-type respectively) was determined for each brain region.

**Figure 4 pathogens-12-00353-f004:**
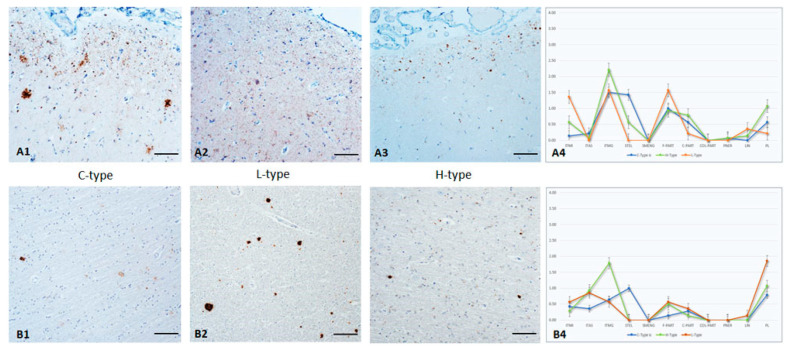
PrP^Sc^ accumulation in cerebral molecular layer (**A1**–**A4**) and white matter (**B1**–**B4**) of C-, L-, and H-type infected cattle (Immunohistochemistry, mab anti-PrP F99). **A1**–**A3**: Cerebral molecular layer in C-type BSE infected animals with a distinct PrP^Sc^ ”tape-like” pattern directly beneath the leptomeninx as well as a multifocal STEL reaction pattern. H-type BSE infected animals show a prominent ITMG reaction pattern, in a comparable “tape-like” appearance. This pattern contradict, L-type BSE which shows a diffuse fine PART distribution of PrP^Sc^, Bar 50 µm; **A4**: PrP^Sc^-profiles in the cerebral molecular layer of all three BSE types; **B1**–**B3**: Cerebral white matter with a distinct PL pattern in L-type BSE, whereas C-type BSE induces a more pronounced STEL and H-type a distinct ITMG reaction pattern, Bar 50µm, **B4**: PrP^Sc^-profiles of all three BSE types in cerebral white matter. The extent of different cellular PrP^Sc^ patterns in each brain region was determined for each animal, and the following scores were used: weak (score 1.0), mild (2.0), moderate (3.0), and severe (4.0). For the graphs, an average value of all animals in each group (C-, H-, and L-type respectively) was determined for each brain region.PL accumulation in the *white matter* of the cerebrum (Figure 4B1–B4) was generally induced by all BSE types, and was detectable for C- and L-type BSE even in mildly affected animals. However, this pattern was most prominent in L-type BSE cases, which showed PL more frequently and in larger sizes. A corresponding histopathological alteration was not seen. In H-type infected animals almost all glial cells were affected by intraglial PrP^Sc^ accumulation, whereas in L- and C-type BSE this reaction pattern was not prominent. A characteristic for C-type BSE was the induction of a STEL reaction pattern, which was never identified in atypical BSE cases. PART PrP^Sc^ accumulations, as well as scattered ITNR staining reactions, were seen independently of the BSE type, which was mostly associated with advanced cases.

**Table 1 pathogens-12-00353-t001:** Details of cattle intracerebrally inoculated at the FLI/Germany and CFIA/Canada with C-, H- and L-type BSE.

BSE	Animal ID	Incubation Time (mpi)	Lesions (H&E)	Obex/PrP^Sc^ Deposition	Age at Inoculation	Inoculation	Site of Inoculation
C-type (CFIA)	25015	26	++	+++	2–3 months	1 mL of a 10% brain homogenate	Midbrain
	25022	27	++	+++			
	25023	27	++	+++			
	25032	26	+	+++			
	25034	24	++	+++			
	29024	20	+	++	5–6 months	1 mL of a 10% brain homogenate	Midbrain
	29026	18	(+)	++			
H-type (CFIA)	29018	18	+	++			
	29033	17	+	++			
L-type (CFIA)	29012	18	++	++			
	29030	17	++	+			
H-type (FLI)	RA10	12	(+)	+	6 months	1 mL of a 10% brain homogenate	Rostral midbrain
	RA13	15	++	+++			
	RA14	14	+	++			
	RA15	16	+	+++			
	RA16	16	++	+++			
L-type (FLI)	RA02	17	+++	+++	6 months	1 mL of a 10% brain homogenate	Rostral midbrain
	RA03	16	++	++			
	RA04	16	+++	+++			
	RA05	11	(+)	(+)			
	RA06	14	++	+			
C-type (FLI-oral)	IT18	50	+++	+++	4–6 months	100 g BSE brainstem homogenate	Oral
	IT23	36	++	+++			
	IT49	36	+++	+++			

(+) = weak; + = mild; ++ = moderate; +++ = severe accumulation of the pathological prion protein (PrP^Sc^); mpi = months post inoculation.

**Table 2 pathogens-12-00353-t002:** Overview of the brain regions examined.

Brain Area	Brain Regions Examined
Brain stem (Obex)	*Hypoglossal Nc. (1)*, DMNV (2), Nc. of the solitary tract (3), Nc. of the spinal tract of the trigeminal nerve (4), Reticular formation (5)
Cerebellum	*Molecular layer (6)*, *Granular layer (7)*, *White matter (8)*, Deep cerebellar Ncc. (9)
Mid brain	*Red Nc. (10)*
Hippocampus	*Hilus (11)*, Str. oriens (12), Str. pyramidale (13), Str. radiatum (14)
Frontal Cortex	*Molecular layer (15a)*, External layer (15b), Internal layer (15c), *White matter (16)*
Telencephalon	*Septal area (17)*

DMNV = Dorsal motor nucleus of the vagus nerve; Nc. = Nucleus; Str. = Stratum; the numbers in brackets indicate the abbreviation used in the graphs; Regions of special interest for BSE-type discrimination are indicated in bold and Italics.

**Table 3 pathogens-12-00353-t003:** Main characteristics in the PrP^Sc^ profiles of C-, H-, and L-type BSE.

Brain Region	Profile	C-Type BSE	H-Type BSE	L-Type BSE
Brainstem/Hypoglossal Nucleus	ITNR	X	X	--
	ITMG	X	XX	(x)
	PART	Fine to coarse	Fine	Fine
	STEL	X	--	--
	LIN	X	(x)	--
Cerebellum/Molecular Layer	ITMG	X	XX	X
	PART	Fine to coarse	Fine	Fine
	STEL	X	(x)	--
	LIN	(x)	(x)	X
	Remarks	Multifocal	Multifocal	Diffuse
Cerebellum/Granular Layer	PNER	--	--	X
	STEL	X	--	--
	Remarks	Multifocal	Multifocal	Diffuse
Cerebellum/White Matter	ITMG	X	XX *	X
	PART	Fine	Fine	Fine
	Remarks	Weakly affected	Distinctly involved	Distinctly involved
Midbrain/Red Nucleus	ITNR	X	X	X
	ITMG	X	XX	X
	PART	Coarse	Fine	Fine
	LIN	X	(x)	(x)
Hippocampus/Hilus	ITNR	X	X	X
	ITMG	X	XX	X
	PART	Mostly coarse	Fine	Fine
	STEL	X	--	--
	PL	X	--	--
	Remarks	Central parts	Peripheral parts	Peripheral parts
Hippocampus/in general	PrP^Sc^	*Str. radiatum* and preference for Hilus	*Str. radiatum* and preference for *Str. pyramidale*	*Str. radiatum* and preference for *Str. pyramidale*
Septal area	ITNR	X	X	X
	ITMG	X	XX	X
	PART	Fine to coarse	Fine	Fine
	STEL	X	--	--
Cerebrum/Molecular Layer	ITNR	(x)	(x)	X
	ITMG	X	XX	X
	PART	Fine to coarse	Fine to coarse	Fine to coarse
	STEL	X	(x)	(x)
	Remarks	PrP^Sc^ tape-like	PrP^Sc^ tape-like	--
Cerebrum/White Matter	ITMG	(x)	XX *	(x)
	PART	Fine to coarse	Fine	Fine to coarse
	STEL	X	--	--
	PL	Small	Small	Frequent and large

ITNR = intraneuronal; ITMG = intramicroglial; PART = particulate; PNER = perineuronal; STEL = stellate; PL = plaque-like; LIN = linear; -- = no PrP^Sc^ detectable; (x) = weak PrP^Sc^ deposition; X = prominent PrP^Sc^ deposition; XX = very characteristic ITMG pattern for the H-type; * involving almost all glial cells.

## Data Availability

The data that support the findings of this study are available in the main manuscript and in the Appendix A for this article.

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
