# Peer review of "Discrimination of Classical and Atypical BSE by a Distinct Immunohistochemical PrPSc Profile"

_pathogens, 2023, doi:10.3390/pathogens12020353_

Round 1

Reviewer 1 Report

This is a comprehensive and detailed comparison of the histopathological and immunohistochemical phenotype of experimentally BSE (C, H and L) infected cattle, with the prurpose to identify BSE strain specific signatures, that could help in discriminatory testing, when fresh-frozen tissues samples are not available, but only FFPE material. For BSE diagnostics this work is of fundamental interest in view of the consequences of confirmed C-type BSE cases on international trade, according to WOAH guidelines.

The experiments are well done and the data are interpreted appropriately. I have no further comments apart that I congratulate the authors for this important work. 

Author Response

Thank you very much for the kind words, we highly appreciate it!

Reviewer 2 Report

See attached file

Author Response

Dear Editor and Reviewer 2,

attached is a document outlining how we have implemented the reviewer's comments and suggestions for improvement.

Sincerely yours

Christine Fast

Reviewer 3 Report

Summary

In this article, the authors present the result of histopathology and immunohistochemistry analysis from 24 brains of cattle experimentally infected by three types of BSE: classical (C-type) and atypical (L- and H-type). For comparison with these 24 intracerebrally inoculated animals authors included 3 cattle orally infected with C-type BSE. Six brain areas with 17 neuroanatomical region have been examined and PrPsc deposition types have been classified in 11 different patterns. The aim of this study is to propose an alternative BSE strain typing test founded on formalin fixed tissue and particularly by the analyze of localization and morphology of PrPsc deposites in brain. In this way, authors shows a well-documented description of the lesion profile and the PrPsc profiles which represent an important contribution for physiopathology of TSE in cattle. These results are based on a consequent experimental group often difficult to carry out with cattle. The major contribution of this study is the description of PrPsc deposition profiles immunolabelled with the antibody F99/97.6.1 raised against the core region of protein. The author present immunohistochemical PrPsc profile results that allow for the differentiation of the three types of BSE in experimentally infected cattle.

General concept comments

As already described by Jeffrey and Gonzalez (2007) for scrapie in sheep, the authors present the description of PrPsc reaction patterns identified in cattle experimentally infected with three types of BSE. The weakness of this work start with the noticeable number of individuals in experimental group rarely carried out for bovines. The description of PrPsc deposition profiles categorized in 11 types in numerous brain areas represent a clear contribution for the field of TSE studies. One of the proposition of the authors is to use this method as an additional diagnostic tool to support the differentiation of BSE types. It is difficult to validate this proposal without the presentation of similar results obtained from field cases. For example, the study of the Hypoglossal Nucleus identified here as a region of interest and located in the brainstem already collected in the context of TSE diagnostics would be very helpful. This observation is supported by the results of this study which describe the variability of the profiles between animals orally and intracerebrally infected with BSE-C identified as a very univocal strain on the one hand and on the other hand the differences in the levels of accumulation of PrPsc in the animals of the study. Moreover, in their study of scrapie, Jeffrey and Gonzalez have, in addition to the morphology and cellular localization of PrPsc deposits, performed an epitopic mapping of PrPsc déposits which, in the context of the characterization of atypical forms of BSE, appears to be of great interest, especially in view of the discriminating results obtained by Western Blot mapping approaches. In situ studies of the resistance of PrPsc deposits to chemical or protein denaturants would also contribute.

Specific comments

Line 128 : the article reference 48 proposed correspond to preliminary description of the experiment, the detailed of this experiment is described here : Balkema-Buschmann et al. J. Toxicol. Environ. Health A, 74 (2011), pp. 103-109. This reference should be more accurate or complementary.

$ 2.2 (lane 141 to 165) : for the experiment carried out in Canada, the description of inoculum is not sufficient : for one type of BSE, was the same inoculum used?

$ 2.5 (lane 197 to 203): the 4 lesions grading defined in the article have to be link with numeric score attributed in the figure showing lesion profiles results. Moreover, the lesion profiles were presented as a mean but standard deviation is not showed, in these conditions it’s difficult to appreciate the results especially given the large number of individuals used in this study.

Lane 214: the article reference (Jeffrey and Gonzalez 2007) is not reference with a number as other references in the text.

Lanes 223 to 224: as for lesion profiles, the grading of PrPsc depositions: weak, mild, moderate and severe have to be link with numeric score attributed in the corresponding figures. In addition, Standard Deviation should be very contributive here too.

Lane 248: it was mentioned “brain areas”, it could be more comprehensive for the reader to use the same designation mentioned previously in table 2: brain regions.

Lane 266: “advanced cases” is mentioned here and elsewhere (lane 298), this term must be defined here.

Lane 354: it was indicated that in the Hippocampus Str. Radiatum all BSE-types were affected but the table 3 did not mentioned it for classical BSE.

Lanes 439 – 440: This study represents an important contribution to the knowledge on the differential characterization of PrPsc deposits in the brain of animals with different types of BSE. However, to claim that this experimental design reflects natural disease without direct comparison with field cases is a bit risky.

Lane 446: 17 instead of 18

Table 1: in the table, the incubation time indicated for the animal RA05 is 12 mpi, but in the previous article (Balkema-Buschmann et al. J. Toxicol. Environ. Health A, 74 (2011)) it was only 11 mpi. Please clarify

Table 2: highlight the nine brain regions which shows differences in PrPsc profiles between BSE types should be more comprehensive for the reader.

Table 3: authors identified 9 brain regions contributive for BSE types differentiation but this table present 10 regions with Hippocampus Str. Radiatum in addition. Maybe it could be integrate in result figure with comparative pictures. Different symbols (X, (x),--) used in the table are not defined.

All figures with PrPsc deposits graphic presentation mentioned INTR instead of ITNR for intraneuronal deposits as defined in $2.6 and intraglial is summarize as ITMGL in this same $2.6 but in figures it’s ITMG.

Figure 1: does not facilitate the understanding of the study because it is different from the organization of the following figures. If the objective of this figure were to present a characteristic image of each type of BSE, presenting three images would be more understandable. As for the iconography of the Red Nucleus, it should follow the same pattern as the presentation of images of the other brain regions of interest.

Figure 2: specified for section B4 if it correspond to granular or molecular layer.

Figure S1: For brain region number 15, an illustration box as for hippocampus should be helpful to present the 3 sub regions.

Figure S2: in the study submeningeal (SMENG) was not observed then it is not necessary to show picture but intra-astrocytic (not intrastroglial as in the text) (ITAS) and perineuronal (PNER) were rare but observed in the study picture of them may be helpful.

Figure S4: the Y-axis value (0 to 16) are not defined.

Table S2: abbreviations should be homogenize between text and table: stellate PrPsc deposits were defined as STEL in $2.6 but as STELL in table S2, for intraglial it’s ITGL in table S2 and ITMGL in $2.6.

Author Response

Dear Editor and Reviewer 3

Attached is a document outlining how we have implemented the reviewer's comments and suggestions for improvement.

Yous sincerely

Christine Fast
